# Learning Character-level Compositionality with Visual Features

## Abstract

Previous work has modeled the compositionality of words by creating character-level models of meaning, reducing problems of sparsity for rare words. However, in many writing systems compositionality has an effect even on the character-level: the meaning of a character is derived by the sum of its parts. In this paper, we model this effect by creating embeddings for characters based on their visual characteristics, creating an image for the character and running it through a convolutional neural network to produce a visual character embedding. Experiments on a text classification task demonstrate that such model allows for better processing of instances with rare characters in languages such as Chinese, Japanese, and Korean. Additionally, qualitative analyses demonstrate that our proposed model learns to focus on the parts of characters that carry categorical content which resulting in embeddings that are coherent in visual space.

## 1 Introduction

Compositionality—the fact that the meaning of larger linguistic units is created by combining the meaning of smaller units—is a hallmark of every natural language (Szabó, 2010). Recently, neural models have provided a powerful tool for learning how to compose words together into a meaning representation of whole sentences for many downstream tasks. This is done using models of various levels of sophistication, from simpler bag-of-words (Iyyer et al., 2015) and linear recurrent neural network (RNN) models (Sutskever et al., 2014; Kiros et al., 2015), to more sophisticated models

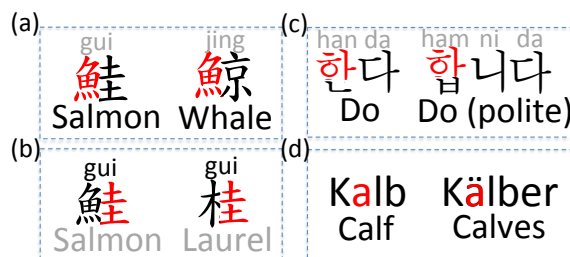

Figure 1: Examples of character-level compositionality in (a, b) Chinese, (c) Korean, and (d) German. The red part of the characters are shared, and affects the pronunciation (top) or meaning (bottom).

using tree-structured (Socher et al., 2013) or convolutional networks (Kalchbrenner et al., 2014).

In fact, a growing body of evidence shows that it is essential to look below the word-level and consider compositionality within words themselves. For example, several works have proposed models that represent words by composing together the characters into a representation of the word itself (Ling et al., 2015; Zhang et al., 2015; Dhingra et al., 2016). Additionally, for languages with productive word formation (such as agglutination and compounding), models calculating morphology-sensitive word representations have been found effective (Luong et al., 2013; Botha and Blunsom, 2014). These models help to learn more robust representations for *rare words* by exploiting morphological patterns, as opposed to models that operate purely on the lexical level as the atomic units.

For many languages, compositionality stops at the character-level: characters are atomic units of meaning or pronunciation in the language, and no further decomposition can be done.[1] However, for other languages, character-level compositionality, where a character's meaning or pronunciation

---

[1] In English, for example, this is largely the case.

| Lang | Geography | Sports | Arts | Military | Economics | Transportation |
|------|-----------|--------|------|----------|-----------|----------------|
| Chinese | 32.4k | 49.8k | 50.4k | 3.6k | 82.5k | 40.4k |
| Japanese | 18.6k | 82.7k | 84.1k | 81.6k | 80.9k | 91.8k |
| Korean | 6k | 580 | 5.74k | 840 | 5.78k | 1.68k |
| Lang | Medical | Education | Food | Religion | Agriculture | Electronics |
| Chinese | 30.3k | 66.2k | 554 | 66.9k | 89.5k | 80.5k |
| Japanese | 66.5k | 86.7k | 20.2k | 98.1k | 97.4k | 1.08k |
| Korean | 16.1k | 4.71k | 33 | 2.60k | 1.51k | 1.03k |

Table 1: By-category statistics for the Wikipedia dataset. Note that Food is the abbreviation for "Food and Culture" and Religion is the abbreviation for "Religion and Belief".

can be derived from the sum of its parts, is very much a reality. Perhaps the most compelling example of compositionality of sub-character units can be found in logographic writing systems such as the Han and Kanji characters used in Chinese and Japanese, respectively.[2] As shown on the left side of Fig. 1, each part of a Chinese character (called a "radical") potentially contributes to the meaning (i.e., Fig. 1(a)) or pronunciation (i.e., Fig. 1(b)) of the overall character. This is similar to how English characters combine into the meaning or pronunciation of an English word. Even in languages with phonemic orthographies, where each character corresponds to a pronunciation instead of a meaning, there are cases where composition occurs. Fig. 1(c) and (d) show the examples of Korean and German, respectively, where morphological inflection can cause single characters to make changes where some but not all of the component parts are shared.

In this paper, we investigate the feasibility of modeling the *compositionality of characters* in a way similar to how humans do: by visually observing the character and using the features of its shape to learn a representation encoding its meaning. Our method is relatively simple, and generalizable to a wide variety of languages: we first transform each character from its Unicode representation to a rendering of its shape as an image, then calculate a representation of the image using Convolutional Neural Networks (CNNs) (Cun et al., 1990). These features then serve as inputs to a down-stream processing task and trained in an end-to-end manner, which first calculates a loss function, then back-propagates the loss back to the CNN.

---

[2]Other prominent examples are largely for extinct languages: Egyptian hieroglyphics, Mayan glyphs, and Sumerian cuneiform scripts (Daniels and Bright, 1996).

As demonstrated by our motivating examples in Fig. 1, in logographic languages, character-level semantic or phonetic similarity is often indicated by visual cues; we conjecture that CNNs can appropriately model these visual patterns. Consequently, characters with similar visual appearances will be biased to have similar embeddings, allowing our model to handle *rare characters* effectively, just as character-level models have been effective for rare words.

To evaluate our model's ability to learn representations, particularly for rare characters, we perform experiments on a downstream task of classifying Wikipedia titles for three Asian languages: Chinese, Japanese, and Korean. We show that our proposed framework outperforms baseline model that use standard character embeddings for instances containing rare characters. A qualitative analysis of the characteristics of the learned embeddings of our model demonstrates that visually similar characters share similar embeddings. We also show that the learned representations are particularly effective under low-resource scenarios and *complementary* with standard character embeddings; combining the two representations through three different fusion methods (Snoek et al., 2005; Karpathy et al., 2014) leads to consistent improvements over the strongest baseline without visual features.

## 2 Dataset

Before delving into the details of our model, we first describe a dataset we constructed to examine the ability of our model to capture the compositional characteristics of characters. Specifically, the dataset must satisfy two desiderata: (1) it must be necessary to fully utilize each character in the input in order to achieve high accuracy, and (2) there must be enough regularity and com-

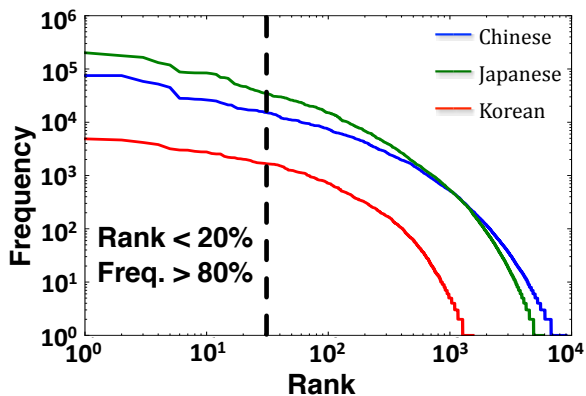

Figure 2: The character rank-frequency distribution of the corpora we considered in this paper. All three languages have a long-tail distribution.

positionality in the characters of the language. To satisfy these desiderata, we create a text classification dataset where the input is a Wikipedia article title in Chinese, Japanese, or Korean, and the output is the category to which the article belongs.[3] This satisfies (1), because Wikipedia titles are short and thus each character in the title will be important to our decision about its category. It also satisfies (2), because Chinese, Japanese, and Korean have writing systems with large numbers of characters that decompose regularly as shown in Fig. 1. While this task in itself is novel, it is similar to previous work in named entity type inference using Wikipedia (Toral and Munoz, 2006; Kazama and Torisawa, 2007; Ratinov and Roth, 2009), which has proven useful for downstream named entity recognition systems.

### 2.1 Dataset Collection

As the labels we would like to predict, we use 12 different main categories from the Wikipedia web page: Geography, Sports, Arts, Military, Economics, Transportation, Health Science, Education, Food Culture, Religion and Belief, Agriculture and Electronics. Wikipedia has a hierarchical structure, where each of these main categories has a number of subcategories, and each subcategory has its own subcategories, etc. We traverse this hierarchical structure, adding each main category tag to all of its descendants in this subcategory tree structure. In the case that a particular article is the descendant of multiple main categories, we favor the main category that minimizes the depth of the

---

[3]This dataset and the crawling scripts are attached as supplementary material, and will be released publicly upon acceptance of the paper.

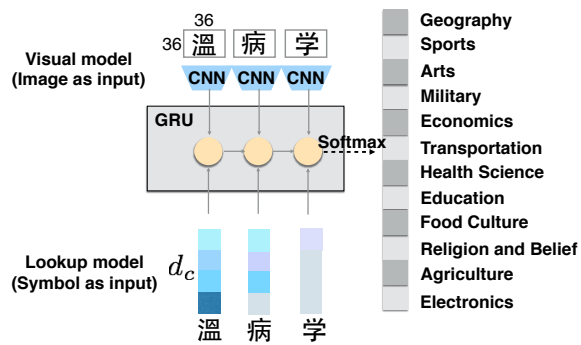

Figure 3: An illustration of two models, our proposed VISUAL model at the top and the baseline LOOKUP model at the bottom using the same RNN architecture. A string of characters (e.g. "温病学"), each converted into a 36x36 image, serves as input of our VISUAL model. $d_c$ is the dimension of the character embedding for the LOOKUP model.

article in the tree (e.g., if an article is two steps away from Sports and three steps away from Arts, it will receive the "Sports" label). We also perform some rudimentary filtering, removing pages that match the regular expression ".*:.*", which catches special pages such as "title:agriculture".

### 2.2 Statistics

For Chinese, Japanese, and Korean, respectively, the number of articles is 593k/810k/46.6k, and the average length and standard deviation of the title is 6.25±3.96/8.60±5.58/6.10±3.71. As shown in Fig. 2, the character rank-frequency distributions of all three languages follows the 80/20 rule (Newman, 2005) (i.e., top 20% ranked characters that appear more than 80% of total frequencies), demonstrating that the characters in these languages belong to a long tail distribution.

We further split the dataset into training, validation, and testing sets with a 6:2:2 ratio. The category distribution for each language can be seen in Tab. 1. Chinese has two varieties of characters, traditional and simplified, and the dataset is a mix of the two. Hence, we transform this dataset into two separate sets, one completely simplified and the other completely traditional using the Chinese text converter provided with Mac OS.

## 3 Model

Our overall model for the classification task follows the encoder model by Sutskever et al. (2014). We calculate character representations, use a RNN

| Layer# | 3-layer CNN Configuration |
|--------|---------------------------|
| 1 | Spatial Convolution $(3, 3) \rightarrow 32$ |
| 2 | ReLu |
| 3 | MaxPool $(2, 2)$ |
| 4 | Spatial Convolution $(3, 3) \rightarrow 32$ |
| 5 | ReLu |
| 6 | MaxPool $(2, 2)$ |
| 7 | Spatial Convolution $(3, 3) \rightarrow 32$ |
| 8 | ReLu |
| 9 | Linear $(800, 128)$ |
| 10 | ReLu |
| 11 | Linear $(128, 128)$ |
| 12 | ReLu |

Table 2: Architecture of the CNN used in the experiments. All the convolutional layers have 32 $3 \times 3$ filters.

to combine the character representations into a sentence representation, and then add a softmax layer after that to predict the probability for each class. As shown in Fig. 3, the baseline model, which we call it the LOOKUP model, calculates the representation for each character by looking it up in a character embedding matrix. Our proposed model, the VISUAL model instead learns the representation of each character from its visual appearance via CNN.

**LOOKUP model**   Given a character vocabulary $C$, for the LOOKUP model as in the bottom part of Fig. 3, the input to the network is a stream of characters $c_1, c_2, ...c_N$, where $c_n \in C$. Each character is represented by a 1-of-$|C|$ (one-hot) encoding. This one-hot vector is then multiplied by the lookup matrix $T_C \in \mathbf{R}^{|C| \times d_c}$, where $d_c$ is the dimension of the character embedding.

**VISUAL model**   The proposed method aims to learn a representation that includes image information, allowing for better parameter sharing among characters, particularly characters that are less common. Different from the LOOKUP model, each character is first transformed into a 36-by-36 image based on its Unicode encoding as shown in the upper part of Fig 3. We then pass the image through a CNN to get the embedding for the image. The parameters for the CNN are learned through backpropagation from the classification loss. Because we are training embeddings based on this classification loss, we expect that the CNN will focus on parts of the image that contain semantic information useful for category classification, a hypothe-

sis that we examine in the experiments (see Section 5.5).

In more detail, the specific structure of the CNN that we utilize consists of three convolution layers where each convolution layer is followed by the max pooling and ReLU nonlinear activation layers. The configurations of each layer are listed in Tab. 2. The output vector for the image embeddings also has size $d_c$ which is the same as the LOOKUP model.

**Encoder and Classifier**   For both the LOOKUP and the VISUAL models, we adopt an RNN encoder using Gated Recurrent Units (GRUs) (Chung et al., 2014). Each of the GRU units processes the character embeddings sequentially. At the end of the sequence, the incremental GRU computation results in a hidden state $e$ embedding the sentence. The encoded sentence embedding is passed through a linear layer whose output is the same size as the number of classes. We use a softmax layer to compute the posterior class probabilities:

$$P(y = j|e) = \frac{\exp(w_j^T e + b_j)}{\sum_{i=1}^{L} \exp(w_i^T e + b_j)} \quad (1)$$

To train the model, we use cross-entropy loss between predicted and true targets:

$$J = \frac{1}{B} \sum_{i=1}^{B} \sum_{j=1}^{L} -t_{i,j} \log(p_{i,j}) \quad (2)$$

where $t_{i,j} \in \{0, 1\}$ represents the ground truth label of the $j$-th class in the $i$-th Wikipedia page title. $B$ is the batch size and $L$ is the number of categories.

## 4   Fusion-based Models

One thing to note is that the LOOKUP and the VISUAL models have their own advantages. The LOOKUP model learns embedding that captures the semantics of each character symbol without sharing information with each other. On the contrary, the proposed VISUAL model directly learns embedding from visual information, which naturally shares information between visually similar characters. This characteristic gives the VISUAL model the ability to generalize better to rare characters, but also has the potential disadvantage of introducing noise for characters with similar appearances but different meanings.

With the complementary nature of these two models in mind, we further combine the two embeddings to achieve better performances. We adopt

| Lookup/Visual | 100% | 50% | 12.5% |
|---|---|---|---|
| zh_trad | 0.55/0.54 | 0.53/0.50 | 0.48/0.47 |
| zh_simp | 0.55/0.54 | 0.53/0.52 | 0.48/0.46 |
| ja | 0.42/0.39 | 0.47/0.45 | 0.44/0.41 |
| ko | 0.47/0.42 | 0.44/0.39 | 0.37/0.36 |

Table 3: The classification results of the Lookup / Visual models for different percentages of full training size.

| Lookup/Visual | 100 | 1000 | 10000 |
|---|---|---|---|
| zh_trad | 0.22/**0.49** | 0.35/0.35 | **0.40**/0.39 |
| zh_simp | 0.25/**0.53** | **0.39**/0.37 | **0.41**/0.40 |
| ja | 0.30/**0.35** | **0.45**/0.41 | **0.44**/0.41 |
| ko | **0.44**/0.33 | **0.44**/0.33 | **0.48**/0.42 |

Table 4: Classification results for the Lookup / Visual of the $k$ lowest frequency instances across four datasets.

three fusion schemes, early fusion, late fusion (described by Snoek et al. (2005) and Karpathy et al. (2014)), and fallback fusion, a method specific to this paper.

**Early Fusion** Early fusion works by concatenating the two varieties of embeddings before feeding them into the RNN. In order to ensure that the dimensions of the RNN are the same after concatenation, the concatenated vector is fed through a hidden layer to reduce the size from $2 \times d_c$ to $d_c$. The whole model is then fine-tuned with training data.

**Late Fusion** Instead of learning a joint representation like early fusion, late fusion averages the model predictions. Specifically, it takes the output of the softmax layers from both models and averages the probabilities to create a final distribution used to make the prediction.

**Fallback Fusion** Our final fallback fusion method hypothesizes that our Visual model does better with instances which contain more rare characters. First, in order to quantify the overall rareness of an instance consisting of multiple characters, we calculate the average training set frequency of the characters therein. The fallback fusion method uses the Visual model to predict testing instances with average character frequency below or equal to a threshold (here we use 0.0 frequency as cutoff, which means all characters in the instance do not appear in the training set), and uses the Lookup model to predict the rest of the instances.

## 5 Experiments and Results

In this section, we compare our proposed Visual model with the baseline Lookup model through three different sets of experiments. First, we examine whether our model is capable of classifying text and achieving similar performance as the baseline model. Next, we examine the hypothesis that our model will outperform the baseline model

when dealing with low frequency characters. Finally, we examine the fusion methods described in Section 4.

### 5.1 Experimental Configurations

The dimension of the embeddings and batch size for both models are set to $d_c = 128$ and $B = 400$, respectively. We build our proposed model using Torch (Collobert et al., 2002), and use Adam (Kingma and Ba, 2014) with a learning rate $\eta = 0.001$ for stochastic optimization. The length of each instance is cut off or padded to 10 characters for batch training.

### 5.2 Comparison with the baseline model

In this experiment, we examine whether our Visual model achieves similar performance with the baseline Lookup model in classification accuracy.

The results in Tab. 3 show that the baseline model performs 1-2% better across four datasets; this is due to the fact that the Lookup model can directly learn character embeddings that capture the semantics of each character symbol for frequent characters. On the contrary, the Visual model learns embeddings from visual information, which constraints characters that has similar appearance to have similar embeddings. This is an advantage for rare characters, but a disadvantage for high frequency characters because being similar in appearance does not always lead to similar semantics.

To demonstrate that this is in fact the case, besides looking at the overall classification accuracy, we also examine the performance on classifying low frequency instances which are sorted according to the average training set frequency of the characters therein. Tab. 4 and Fig. 4 (blue lines) both show that our model performs better in the 100 lowest frequency instances (the intersection point of the two models); this confirms that the Visual model can share visual information among characters and help to classify low frequency instances.

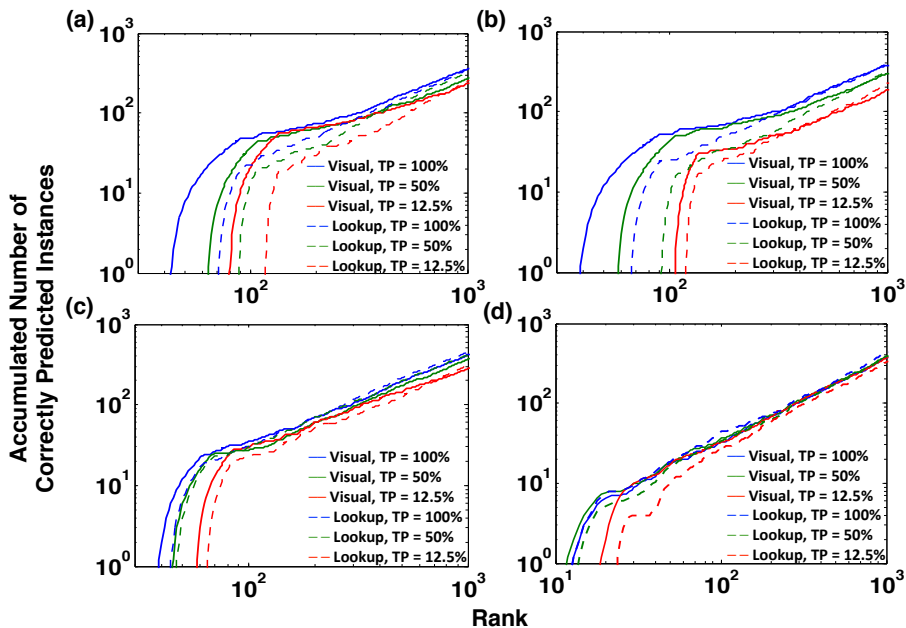

Figure 4: Experiments on different training sizes for four different datasets. More specifically, we consider three different training data size percentages (TPs) (100%, 50%, and 12.5%) and four datasets: (a) traditional Chinese, (b) simplified Chinese, (c) Japanese, and (d) Korean. We calculate the accumulated number of correctly predicted instances for the VISUAL model (solid lines) and the LOOKUP model (dashed lines).

## 5.3 Experiments on different training data sizes

In our second experiment, we consider two smaller training sizes (i.e., 50% and 12.5% of the full training size) indicated by green and red lines in Fig. 4. We performed this experiment under the hypothesis that because the proposed method was more robust to infrequent characters, the proposed model may perform better in low-resourced scenarios. If this is the case, the intersection point of the two models will shift right because of the increase of the number of instances with low average character frequency.

As we can see in Fig. 4, the intersection point for 100% training data lies between the intersection point for 50% training data and 12.5%. This disagrees with our hypothesis; this is likely because while the number of low-frequency characters increases, smaller amounts of data also adversely impact the ability of CNN to learn useful visual features, and thus there is not a clear gain nor loss when using the proposed method.

As a more extreme test of the ability of our proposed framework to deal with the unseen characters in the test set, we use traditional Chinese as our training data and simplified Chinese as our testing

|         | zh_trad | zh_simp | ja     | ko     |
|---------|---------|---------|--------|--------|
| Lookup  | 0.5503  | 0.5543  | 0.4914 | 0.4765 |
| Visual  | 0.5434  | 0.5403  | 0.4775 | 0.4207 |
| early   | 0.5520  | 0.5546  | 0.4896 | 0.4796 |
| late    | **0.5658** | **0.5685** | **0.5029** | **0.4869** |
| fall    | 0.5507  | 0.5547  | 0.4914 | 0.4766 |

Table 5: Experiment results for three different fusion methods across 4 datasets.

data. The model was able to achieve around 40% classification accuracy when we use the full training set, compared to 55%, which is achieved by the model trained on simplified Chinese. This result demonstrates that the model is able to transfer between similar scripts, similarly to how most Chinese speakers can guess the meaning of the text, even if it is written in the other script.

## 5.4 Experiment on Different Fusion Methods

Results of different fusion methods can be found in Tab. 5. The results show that late fusion gives the best performance among all the fusion schemes combining the LOOKUP model and the proposed VISUAL model. Early fusion achieves small improvements for all languages except Japanese, where it displays a slight drop. Unsurprisingly,

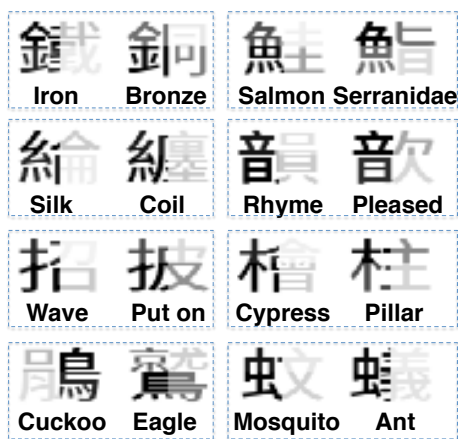

Figure 5: Examples of how much each part of the character contributes to its embedding (the darker the more). Two characters are shown per radical to emphasize that characters with same radical have similar patterns.

fallback fusion performs better than the LOOKUP model and the VISUAL model alone, since it directly targets the weakness of the LOOKUP model (e.g., rare characters) and replaces the results with the VISUAL model. These results show that simple integration, no matter which schemes we use, is beneficial, demonstrating that both methods are capturing complementary information.

### 5.5 Visualization of Character Embeddings

Finally, we qualitatively examine what is learned by our proposed model in two ways. First, we visualize which parts of the image are most important to the VISUAL model's embedding calculation. Second, we show the 6-nearest neighbor results for characters using both the LOOKUP and the VISUAL embeddings.

**Emphasis of the VISUAL Model** In order to delve deeper into what the VISUAL model has learned, we measure a modified version of the occlusion sensitivity proposed by Zeiler and Fergus (2014) by masking the original character image in four ways, and examine the importance of each part of the character to the model's calculated representations. Specifically, we leave only the upper half, bottom half, left half, or right half of the image, and mask the remainder with white pixels since Chinese characters are usually formed by combining two radicals vertically or horizontally. We run these four images forward through the CNN part of the model and calculate the $L_2$ distance between

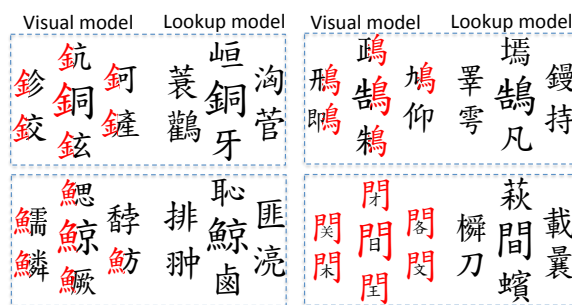

Figure 6: Visualization of the Chinese traditional characters by finding the 6-nearest neighbors of the query (i.e., center) characters. The highlighted red indicates the radical along with the meaning of the characters.

the masked image embeddings with the full image embedding. The larger the distance, the more the masked part of the character contributes to the original embedding. The contribution of each part (e.g. the $L_2$ distance) is represented as a heat map, and then it is normalized to adjust the opacity of the character strokes for better visualization. The value of each corner of the heatmap is calculated by adding the two $L_2$ distances that contribute to this corner.

The visualization is shown in Fig. 5. The meaning of each Chinese character in English is shown below the Chinese character. The opacity of the character strokes represent how much the corresponding parts contribute to the original embedding (the darker the more). In general, the darker part of the character is related to its semantics. For example, "金" means gold in Chinese, which is highlighted in both "鐵" (Iron) and "銅" (Bronze). We can also find similar results for other examples shown in Fig. 5.

**K-nearest neighbors** Finally, to illustrate the difference of the learned embeddings between the two models, we display 6-nearest neighbors ($L_2$ distance) for selected characters in Fig. 6. As can be seen, the VISUAL embedding for characters with similar appearances are close to each other. In addition, similarity in the radical part indicates semantic similarity between the characters. For example, the characters with radical "鳥" all refer to different type of birds.

The LOOKUP embedding do not show such feature, as it learns the embedding individually for each symbol and relies heavily on the training set and the task. In fact, the characters shown in Fig. 6

for the LOOKUP model do not exhibit semantic similarity either. There are two potential explanations for this: First, the category classification task that we utilized do not rely heavily on the fine-grained semantics of each character, and thus the LOOKUP model was able to perform well without exactly capturing the semantics of each character precisely. Second, the Wikipedia dataset contains a large number of names and location and the characters therein might not have the same semantic meaning used in daily vocabulary.

## 6   Related Work

Methods that utilize neural networks to learn distributed representations of words or characters have been widely developed. However, word2vec (Mikolov et al., 2013), for example, requires storing an extremely large table of vectors for all word types. For example, due to the size of word types in twitter tweets, work has been done to generate vector representations of tweets at character-level (Dhingra et al., 2016).

There is also work done in understanding mathematical expressions with a convolutional network for text and layout recognition by using an attention-based neural machine translation system (Deng et al., 2016). They tested on real-world rendered mathematical expressions paired with LaTeX markup and show the system is effective at generating accurate markup. Other than that, there are several works that combine visual information with text in improving machine translation (Sutskever et al., 2014), visual question answering, caption generation (Xu et al., 2015), etc. These works extract image representations from a pre-trained CNN (Zhu et al., 2016; Wang et al., 2016).

Unrelated to images, CNNs have also been used for text classification (Kim, 2014; Zhang et al., 2015). These models look at the sequential dependencies at the word or character-level and achieve the state-of-the-art results. These works inspire us to use CNN to extract features from image and serve as the input to the RNN. Our model is able to directly back-propagate the gradient all the way through the CNN, which generates visual embeddings, in a way such that the embedding can contain both semantic and visual information.

Several techniques for reducing the rare words effects have been introduced in the literature, including spelling expansion (Habash, 2008), dictionary term expansion (Habash, 2008), proper name transliteration (Daumé and Jagarlamudi, 2011), treating words as a sequence of characters (Luong and Manning, 2016), subword units (Sennrich et al., 2015), and reading text as bytes (Gillick et al., 2015). However, most of these techniques still have no mechanism for handling low frequency characters, which are the target of this work.

Finally, there is one work by Shi et al. (2015) on "radical embedding", which explicitly splits Chinese characters into radicals based on a dictionary of what radicals are included in which characters. The motivation of this method is similar to ours, but is only applicable to Chinese, in contrast to the method in this paper, which works on any language for which we can render text.

## 7   Conclusion and Future Work

In this paper, we proposed a new framework that utilizes appearance of characters, convolutional neural networks, recurrent neural networks to learn embeddings that are compositional in the component parts of the characters. More specifically, we collected a Wikipedia dataset, which consists of short titles of three different languages and satisfies the compositionality in the characters of the language. Next, we proposed an end-to-end model that learns visual embeddings for characters using CNN and showed that the features extracted from the CNN include both visual and semantic information. Furthermore, we showed that our VISUAL model outperforms the LOOKUP baseline model in low frequency instances. Additionally, by examining the character embeddings visually, we found that our VISUAL model is able to learn visually related embeddings.

In summary, we tackled the problem of rare characters by using embeddings learned from images. In the future, we hope to further generalize this method to other tasks such as pronunciation estimation, which can take advantage of the fact that pronunciation information is encoded in parts of the characters as demonstrated in Fig. 1, or machine translation, which could benefit from a wholistic view that considers both semantics and pronunciation. We also hope to apply the model to other languages with complicated compositional writing systems, potentially including historical texts such as hieroglyphics or cuneiform.

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
