# Peer review of "Learning Character-level Compositionality with Visual Features"

_ACL 2017 — decision unknown_

[Official Review · Reviewer 1 · rating 4 · confidence 4]
soundness 3 · originality 3 · clarity 2 · impact 3 · substance 4 · appropriateness 5 · meaningful comparison 3 · presentation format Oral Presentation

- Update after rebuttal

I appreciate the authors taking the time to clarify their implementation of the
baseline and to provide some evidence of the significance of the improvements
they report. These clarifications should definitely be included in the
camera-ready version. I very much like the idea of using visual features for
these languages, and I am looking forward to seeing how they help more
difficult tasks in future work.

- Strengths:

- Thinking about Chinese/Japanese/Korean characters visually is a great idea!

- Weaknesses:

- Experimental results show only incremental improvement over baseline, and the
choice of evaluation makes it hard to verify one of the central arguments: that
visual features improve performance when processing rare/unseen words.

- Some details about the baseline are missing, which makes it difficult to
interpret the results, and would make it hard to reproduce the work.

- General Discussion:

The paper proposes the use of computer vision techniques (CNNs applied to
images of text) to improve language processing for Chinese, Japanese, and
Korean, languages in which characters themselves might be compositional. The
authors evaluate their model on a simple text-classification task (assigning
Wikipedia page titles to categories). They show that a simple one-hot
representation of the characters outperforms the CNN-based representations, but
that the combination of the visual representations with standard one-hot
encodings performs better than the visual or the one-hot alone. They also
present some evidence that the visual features outperform the one-hot encoding
on rare words, and present some intuitive qualitative results suggesting the
CNN learns good semantic embeddings of the characters.

I think the idea of processing languages like Chinese and Japanese visually is
a great one, and the motivation for this paper makes a lot of sense. However, I
am not entirely convinced by the experimental results. The evaluations are
quite weak, and it is hard to say whether these results are robust or simply
coincidental. I would prefer to see some more rigorous evaluation to make the
paper publication-ready. If the results are statistically significant (if the
authors can indicate this in the author response), I would support accepting
the paper, but ideally, I would prefer to see a different evaluation entirely.

More specific comments below:

- In Section 3, paragraph "lookup model", you never explicitly say which
embeddings you use, or whether they are tuned via backprop the way the visual
embeddings are. You should be more clear about how the baseline was
implemented. If the baseline was not tuned in a task-specific way, but the
visual embeddings were, this is even more concerning since it makes the
performances substantially less comparable.

- I don't entirely understand why you chose to evaluate on classifying
wikipedia page titles. It seems that the only real argument for using the
visual model is its ability to generalize to rare/unseen characters. Why not
focus on this task directly? E.g. what about evaluating on machine translation
of OOV words? I agree with you that some languages should be conceptualized
visually, and sub-character composition is important, but the evaluation you
use does not highlight weaknesses of the standard approach, and so it does not
make a good case for why we need the visual features. 

- In Table 5, are these improvements statistically significant?

- It might be my fault, but I found Figure 4 very difficult to understand.
Since this is one of your main results, you probably want to present it more
clearly, so that the contribution of your model is very obvious. As I
understand it, "rank" on the x axis is a measure of how rare the word is (I
think log frequency?), with the rarest word furthest to the left? And since the
visual model intersects the x axis to the left of the lookup model, this means
the visual model was "better" at ranking rare words? Why don't both models
intersect at the same point on the x axis, aren't they being evaluated on the
same set of titles and trained with the same data? In the author response, it
would be helpful if you could summarize the information this figure is supposed
to show, in a more concise way. 

- On the fallback fusion, why not show performance for for different
thresholds? 0 seems to be an edge-case threshold that might not be
representative of the technique more generally.

- The simple/traditional experiment for unseen characters is a nice idea, but
is presented as an afterthought. I would have liked to see more eval in this
direction, i.e. on classifying unseen words

- Maybe add translations to Figure 6, for people who do not speak Chinese?